# Peripheral nervous system involvement associated with COVID-19. A systematic review of literature

**Andreea-Raluca Hanganu**[1,2]*, **Alexandru Constantin**[3,4], **Elena-Sonia Moise**[5], **Cristian-Mihail Niculae**[1,2], **Ioana Diana Olaru**[6,7], **Cristian Băicuş**[1,3], **Adriana Hristea**[1,2]

1 Departament of Infectious Diseasese, Carol Davila University of Medicine and Pharmacy, Bucharest, Romania, 2 Matei Balş National Institute of Infectious Diseases, Bucharest, Romania, 3 Colentina Clinical Hospital, Internal Medicine Department, Bucharest, Romania, 4 Department of Internal Medicine, Carol Davila University of Medicine and Pharmacy, Bucharest, Romania, 5 Department of Pathology, Carol Davila University of Medicine and Pharmacy, Bucharest, Romania, 6 London School of Hygiene & Tropical Medicine, London, United Kingdom, 7 Institute for Medical Microbiology, University Hospital Münster, Münster, Germany

* andreea.florea21@gmail.com

**Data Availability Statement:** All relevant data are within the manuscript and its Supporting Information files.

## Abstract

There is increasing evidence of both central and peripheral nervous system (PNS) involvement in COVID-19. We conducted this systematic literature review to investigate the characteristics, management and outcomes of patients with PNS, including the types and severity of cranial nerves (CN) involvement. We systematically searched on PubMed for studies reporting adult patients diagnosed with COVID-19 and PNS involvement until July 2021. From 1670 records, 225 articles matched the inclusion criteria, with a total of 1320 neurological events, in 1004 patients. There were 805 (61%) CN, 350 (26.5%) PNS, and 165 (12.5%) PNS plus CN events. The most frequently involved CN were the facial, vestibulocochlear and olfactory nerve in 27.3%, 25.4% and 16.1%, respectively. Guillain-Barre syndrome spectrum was identified in 84.2% of PNS events. We analysed 328 patients reported in 225 articles with CN, PNS, and PNS plus CN involvement. The patients with CN involvement were younger (mean age 46.2±17.1, p = .003), and were more frequently treated as outpatients (p < .001), mostly with glucocorticoids (p < .001). Patients that had PNS with or without CN involvement were more likely to be hospitalized (p < .001), and to receive intravenous immunoglobulins (p = .002) or plasma exchange (p = .002). Patients with CN, PNS, and PNS plus CN had severe COVID-19 disease in 24.8%, 37.3%, 34.9% respectively. The most common neurological outcome was mild/moderate sequelae in patients with CN, PNS, and PNS plus CN in 54.7%, 67.5% and 67.8% respectively (p = .1) and no significant difference was found between the three categories regarding death, disease severity, time from disease onset to neurological symptoms, lack of improvement and complete recovery. CN involvement was the most frequent PNS finding. All three categories of PNS involvement were rather associated to non-severe COVID-19 but it may be an important cause of hospitalization and post COVID-19 sequelae.

**Funding:** The author(s) received no specific funding for this work.

**Competing interests:** The authors have declared that no competing interests exist.

## Introduction

SARS-CoV-2 is a respiratory virus spread, but more recent studies show that the primary tropism of SARS-CoV-2 is not limited to the respiratory tract. Furthermore, there are increasing reports of cases with nervous system involvement, suggesting neurotropism of the virus [1]. A large variety of nervous system manifestations have been reported, from headache and fatigue to stroke, encephalopathy and coma [2]. In addition to central nervous system (CNS) manifestations, a broad spectrum of peripheral nervous system (PNS) manifestations has been described [2].

The hallmark of PNS involvement in SARS-CoV-2 infection is the Guillain Barré syndrome (GBS) with its most common subtypes: acute inflammatory demyelinating polyneuropathy (AIDP), acute motor axonal neuropathy (AMAN), acute motor sensory axonal neuropathy (AMSAN), Miller Fisher syndrome (MFS), pharyngeal-cervical-brachial GBS (PCB-GBS). To date most systematic literature reviews on PNS involvement have focused on the GBS spectrum [3]. Given the increasing number of patients with neurological manifestations, this area would deserve more attention.

We conducted this systematic literature review to investigate the characteristics, management and outcomes of patients with PNS involvement in COVID-19, including the GBS spectrum, the CN, mononeuritis, dysautonomia, and other types of PNS manifestations. We structured the review in two parts. The first part describes the peripheral nervous system manifestations associated with COVID-19. In the second part we explored associations between PNS and CN involvement and characteristics of patients with COVID-19. We also aimed to discuss possible mechanisms of the PNS findings.

## Methods

### Study design and methods

We registered the study protocol in the PROSPERO database with number CRD42021262017. We systematically searched published studies reporting cases of adult patients diagnosed with COVID-19 and PNS involvement that matched our inclusion and exclusion criteria. The search terms were used as keywords and in combination as MeSH terms in order to maximize the output from literature findings Two authors (ARH, AC) searched PUBMED up to July 2021 using the keywords below:

(((((covid-19) OR (sars-cov-2)) OR (covid)) OR (covid19)) AND (((((((((((((((((((((((((((((((((((mononeuritis) OR (mononeuritis multiplex)) OR (plexopathy)) OR (polyneuropathy)) OR (polyradiculoneuritis)) OR (miller fisher)) OR (peripheral neuropathy)) OR (neuropathy)) OR (neuritis)) OR (polyneuritis))) OR (cranial neuropathy)) OR (olfactory nerve)) OR (optic nerve)) OR (oculomotor nerve)) OR (trochlear nerve)) OR (trigeminal nerve)) OR (abducens nerve)) OR (facial nerve)) OR (vestibulocochlear nerve)) OR (glossopharyngeal nerve)) OR (vagus nerve)) OR (accessory nerve)) OR (hypoglossal nerve)) OR (diplopia)) OR (eyelid ptosis)) OR (strabismus)) OR (facial hypoesthesia)) OR (facial palsy)) OR (facial paresis)) OR (bell's palsy)) OR (vertigo)) OR (hypoacusis)) OR (dizziness)) OR (dysarthria)) OR (dysphagia)) OR (hypoglossal palsy)) OR (guillain barre))

Using a PICO framework, our target population of interest for this systematic review was adult patients ($\geq$18 years old) diagnosed with COVID-19 (SARS-COV-2 infection confirmed by a RT-PCR or antigen test) who had a new onset of peripheral nervous system manifestations, including CN neuropathies within the six months following the SARS-CoV2 infection, and which were presumed to be associated with COVID-19.

### Studies eligibility criteria and data extraction

**Inclusion criteria.** We used the following inclusion criteria: any English, Spanish or French written articles reporting on at least patient diagnosed with COVID-19 by either a RT-PCR or antigen test that associated a peripheral nervous system or CN neuropathy either at the time of infection or up to 6 months thereafter. We defined and included any of the following peripheral neuropathies: CN neuropathies, motor or sensitive nerve neuropathies, either mononeuropathies or polyneuropathies, Guillain Barré syndrome with all its subtypes and any autonomic nervous system manifestations.

**Exclusion criteria.** Articles were excluded according to the following criteria: articles reporting only on pediatric cases, those on patients in whom the COVID-19 diagnosis was made clinically or using serological tests, articles reporting neuropathies in patients vaccinated for SARS-CoV-2 within the previous three months in whom the neurological findings could have been attributed to vaccination, articles reporting on patients only with anosmia and/or ageusia that were not clearly stated to be caused by direct nerve involvement, and articles reporting cases of preexisting neuropathies secondary to other causes.

Study eligibility was established by two authors (AF, AC) who initially evaluated the abstracts based on the eligibility criteria. Disagreements were discussed with two other authors (CB, AH). Duplicated reports were excluded. Full text screening was conducted in duplicate based on inclusion and exclusion criteria. Data were abstracted in duplicate by two authors (ARH, AC). The following information was obtained from each study: study type, number of patients included, demographics, level of care (intensive care unit (ICU)/general ward /outpatient), COVID-19 severity, time between the onset of COVID-19 symptoms and that of the neurological manifestations, type of neuropathy, type of nerves involved, neuropathy treatment used, neurologic outcomes.

All articles were included in the qualitative synthesis and articles reporting on individual patient data were included in the quantitative synthesis as summarised in the PRISMA flow chart in Fig 1. A detailed checklist of the PRISMA criteria is available in S1 Fig.

### Statistical analysis

The descriptive statistical data for categorical variables were presented as numbers/percentages and continuous variables as medians (range) or means ± standard deviation (SD). We used the Shapiro–Wilk test to assess if our numerical data were normally distributed. Ordinal variables and continuous variables for data that significantly deviated from a normal distribution were compared using the Kruskall-Wallis test for $\geq 3$ groups. We applied post hoc tests in order to compare the variables separately. The t-test was used for normally distributed continuous variables. Categorical variables were compared among groups using the chi-squared test. A p-value of $< .05$ was considered statistically significant. We analyzed the collected data using the Statistical Package for Social Sciences (SPSS version 23, IBM Corp., Armonk, NY, USA).

## Results

We included in our analysis data provided in 225 studies summing a total of 1320 pooled neurological events, in 1004 patients that presented with both CN involvement and PNS manifestations. In *Table 1* we synthesized the data extracted from the case series that did not provide individual data in order to be analyzed.

The gender was known for 910 patients and 513(51.1%) were male. Four hundred and twenty-nine (42.7%) patients had non-severe forms of COVID-19, 133 (13.2%) had severe forms of disease, while in 334 (33.3%) patients the disease form was not reported.

For the neurological outcome we assumed the outcome reported in the original articles.

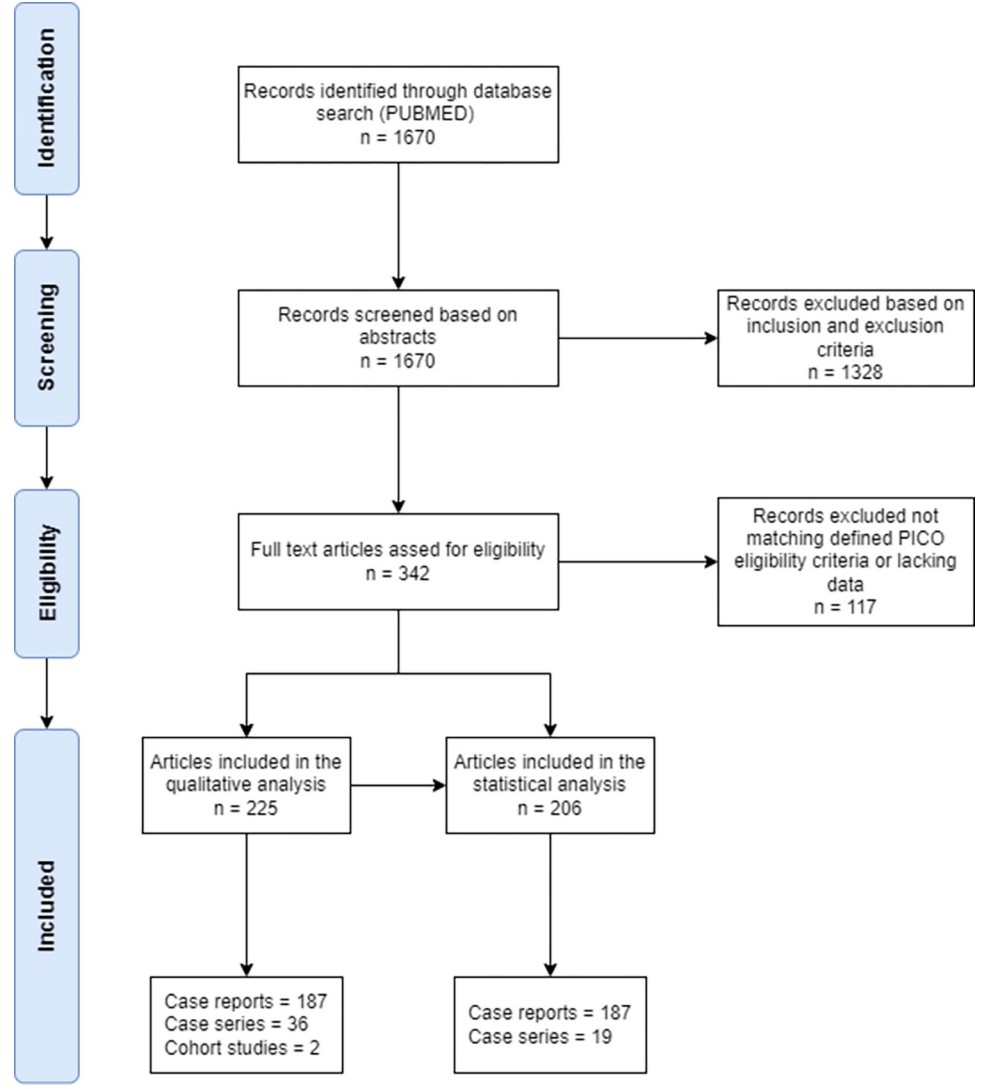

**Fig 1. PRISMA flow diagram showing articles selection strategy.**

Descriptive data of the cranial nerve involvement is presented in *Table 2*.

The facial nerve (VII) was the most frequently involved (27.3%), followed by the vestibulo cochlear nerve, VIII (25.4%) and olphactory, I (16.1%) The least affected by COVID-19 was the hypoglossal nerve, XII (1.1%). In patients with both types of neurological involvement, GBS spectrum was the most common type of PNS involvement, in 147/152 (96.7%) reported cases. Dysautonomia and MNM were only rarely reported. Vestibulo-cochlear nerve (VIII) involvement was most often isolated, in contrast with nerves III (oculomotors), IV (trochlear), VI (abducens), VII (facial), IX (glossopharyngeal) and X (vagus), which were affected also in combination with different types of PNS. The trigeminal nerve (V) was the only CN mostly involved in combination with PNS findings rather than isolated. Even though we did not include patients with isolated anosmia, involvement of the first CN was found in combination with other CN, rather than associated with other PNS findings. From those studies reporting on patients with PNS involvement, unclassified Guillain Barre syndrome (uGBS) was the most

**Table 1. Demographics and characteristics of peripheral nervous system involvement in large case series studies.**

| Nr. | Author | No of patients | Male sex N (%) | COVID disease type | PNS | CN | Diagnosis | Treatment | Neurological outcome |
|---|---|---|---|---|---|---|---|---|---|
| 1 | Yaseen et. al. [4] | 26 | 6 (23) | Non-severe | | VIII 26 | Audiometry MRI | GC | 21 partially improved |
| 2 | Foresti et. al. [5] | 17 | 11 (65) | Severe | AIDP 9 | V/VII 8 | EMG, CSF | 15IVIG, 1PLEX | 15 improved 1 death |
| 3 | Doblan et. al. [6] | 135 | 71 (52.5) | 28 severe | | I 82 II 15 III+IV +VI 19 V 5 VII 93 VIII 52 IX 76 X 28 XI 11 | Clinical | No treat | 37 incomplete recovery 98 complete recovery |
| 4 | Ozdemir et. al. [7] | 33 | 26 (79) | NA | | VII 33 | NA | NA | NA |
| 5 | Altundag et. al. [8] | 24 | 10 (42) | NA | | 24I | NA | NA | NA |
| 6 | Dhamne et. al. [9] | 42 | 31 (74) | 14 severe | AIDP 25 AMSAN 9 AIDP+AMSAN 3 PNP 3 12 GBS | MFS 3 VIII 5 IX+X 1 III+IV +VI 3 | EMG, CSF | 31 IVIG 1 GC 3 no treat | Mild/moderate sequelae |
| 7 | Dharmarajan et. al. [10] | 53 | NA | Non severe | | VIII 53 | Audiometry | NA | NA |
| 8 | Travi et. al. [11] | 6 | NA | NA | GBS 6 | | CSF | NA | NA |
| 9 | Keddie et. al. [12] | 13 | 11 (85) | NA | AIDP 7 GBS 5 | MFS 1 | CSF±EMG | 9 IVIG 4 no treat | 1 death |
| 10 | Hinduja et. al [13] | 13 | 8 (61.5) | NA | Dysautonomia 13 | | FESC/HESC | NA | NA |
| 11 | Delome et. al. [14] | 9 | NA | NA | GBS 5 | VI 1 I 1 III 1 XII 2 | EMG, CSF | NA | NA |
| 12 | Fragiel et. al. [15] | 11 | 6 (54.5) | NA | AIDP 7 AMAN 2 PNP 1 | VII 1 | CSF | NA | 0 mortality |
| 13 | Meppiel et. al. [16] | 19 | 13 (68) | 5 severe | AIDP 15 MNM 1 | III 2 VII 1 | EMG, CSF, MRI | 14 IVIG | 1 mortality 3 full resolution |
| 14 | Tony A et. al. [17] | 107 | 51 (48) | NA | GBS 107 | | NA | NA | NA |
| 15 | Viola et. al. [18] | 77 | 42 (54.5) | NA | | VIII 77 | NA | NA | NA |
| 16 | Rifino et. al. [19] | 17 | 13 (76) | Severe | GBS 17 | 13 V/VIII | NA | NA | NA |
| 17 | Khedr et. al. [20] | 35 | NA | NA | GBS 4 | I 9 VII 2 I+VII 22 | NA | NA | NA |
| 18 | Koh et. al. [21] | 11 | 11 (100) | 2 severe | AMSAN 1 Dysautonomia 5 | VII 5 | EMG, autonomic function tests | GC/PLEX | 2 improved |

(*Continued*)

**Table 1.** (Continued)

| Nr. | Author | No of patients | Male sex N (%) | COVID disease type | PNS | CN | Diagnosis | Treatment | Neurological outcome |
|-----|--------|----------------|----------------|--------------------|-----|-----|-----------|-----------|----------------------|
| 19 | Karadaş et. al. [22] | 18 | NA | NA | GBS 1 | 8 V<br>9 IX | NA | NA | NA |

Abbreviations: F–female, M–male, PNS–peripheral nervous system, CN–cranial nerves, CSF–cerebrospinal fluid, AIDP–acute inflammatory demyelinating polyneuropathy, AMAN–acute motor axonal neuropathy, AMSAN–acute motor and sensitive axonal neuropathy, GBS–Guillain Barre syndrome, MFS—Miller Fisher syndrome, PNP–polyneuropathy, MNM–mononeuritis multiplex, GC—glucocorticoids, PLEX–plasma exchange, IVIG–intravenous immunoglobulins, EMG–electromyography, FESC–feet electrochemical skin conductance, HESC–hand electrochemical skin conductance, I–olfactory nerve, II–optic nerve, III–oculomotor nerve, IV–trochlear nerve, V–trigeminal nerve, VI–abducens nerve, VII–facial nerve, VIII–vestibulocochlear nerve, IX–glossopharyngeal nerve, X–vagus nerve, XI–accessory nerve, XII–hypoglossal nerve, NA–not available.

frequently encountered, in more than 50% of the cases, in both isolated and PNS + CN involvement, followed by AIDP and dysautonomia (*Table 3*).

The pharyngo-cervical-brachial form of GBS was the rarest one as it was found in only one patient [23]. Most cases of GBS were diagnosed using clinical criteria and/or cerebrospinal fluid (CSF) analysis.

Several patients with mononeuritis or mononeuritis multiplex were reported, usually involving the motor part of the sciatic nerve. Other types of peripheral nerve involvement were rare. We identified two cases of post infectious carpal tunnel syndrome, raising concerns about cross reaction between SARS-CoV- 2 antigens and synovial cells [24]. Parsonage Turner was also found in a few patients, appearing in one case during the first days of hospitalization for COVID-19 [25]. Another patient developed the symptoms after four months of persistent dyspnea due to COVID-19, hypothesizing diaphragmatic involvement prior to neuralgic amyotrophia [26]. A few cases of peripheral polyneuropathy that did not meet the GBS criteria for various reasons were identified; one patient responded well to plasma exchange, though, suggesting an immunologic mechanism [27–29]. In one case the neurological symptoms were self-limited but preceded the respiratory symptoms due to COVID-19 by a few days [27].

Here we present the results of the qualitative analysis of the articles reporting individual patient data. Data is shown in *Table 4*. The articles included in this synthesis are available in S1 Table.

**Table 2. Types of cranial nerves involvement in patients with COVID-19.**

| Cranial nerve (CN) | Studies that mention CN involvement | Pooled events | Isolated CN involvement | CN + PNS Involvement |
|--------------------|-------------------------------------|---------------|-------------------------|----------------------|
| | 195 | 957 (%) | 805 (%) | 152 (%) |
| I (olphactory) | 11 | 154 (16.1) | 151 (18.7) | 2 (1.3) I + GBS, 1 I + MNM, 1 |
| II (optic) | 10 | 24 (2.5) | 23 (2.8) | 1 (0.6) II + GBS |
| III + IV + VI (oculomotor, trochlear, abducens) | 29 | 56 (5.8) | 38 (4.7) | 18 (11.8) III + IV + VI + GBS, 17 III + IV + VI + dysautonomia, 1 |
| V(trigeminal) | 8 | 41(4.3) | 15 (1.9 | 26 (17.1) V+ GBS, 25 V+ dysautonomia, 1 |
| VII (facial) | 70 | 261 (27.3) | 190 (23.6) | 71 (46.7) VII+ GBS, 70 VII+ dysautonomia, 1 |
| VIII (vestibulo-cochlear) | 25 | 243 (25.4) | 241 (29.7) | 2 (1.3) VIII+ GBS, 1 VIII+ dysautonomia, 1 |
| IX (glossopharingeal), | 16 | 99 (10.3) | 86 (10.7) | 13 (8.5) IX + GBS |
| X (vagus) | 17 | 49 (5.1) | 35 (4.3) | 14 (9.2) X + GBS |
| XI (accessory) | 3 | 20 (2.1) | 17 (2.1) | 3 (1.9) XI + GBS |
| XII (hypoglossal) | 6 | 11 (1.1) | 9 (1.1) | 2 (1.3) XII + GBS |

Abbreviations: N–number, CN–cranial nerves, PNS–peripheral nervous system, GBS–Guillain Barre Syndrome, MNM–mononeuritis multiplex

**Table 3. Type of PNS involvement in patients with COVID-19.**

| Type of PNS involvement | Studies | Pooled events | Isolated PNS involvement N(%) | PNS + CN involvement N(%) |
|---|---|---|---|---|
| | **183** | **515(%)** | **350 (68)** | **165 (32)** |
| uGBS, | 75 | 273 (53) | 183 (52.3) | 90 (54.5) |
| AIDP | 37 | 99 (19.2) | 62 (17.7) | 38 (10.8) |
| AMAN | 13 | 13 (2.5) | 9 (2.6) | 4 (1.1) |
| AMSAN | 19 | 31 (5.6) | 25 (7.1) | 6 (1.7) |
| MFS | 13 | 18 (3.5) | - | 18 (5.1) |
| MNM | 5 | 14 (2.7) | 11 (3.1) | 3 (1.8) |
| Dysautonomia | 8 | 56 (10.9) | 52 (14.8) | 4 (2.4) |
| Others | 13 | 10 (1.9) | 8 (2.3) | 2 (1.2) |

Abbreviations: uGBS–unclassified Guillain Barre Syndrome, acute inflammatory demyelinating polyneuropathy, AMAN–acute motor axonal neuropathy, AMSAN–acute motor and sensitive axonal neuropathy, MFS–Miller Fischer Syndrome, MNM–mononeuritis multiplex.

Patients with CN involvement were younger (mean of 46.2 ±17.1 years, p = .003) than those with either isolated PNS involvement or both types of neurological involvement (mean of 52.9 ±15.1 and 51.6±15). In terms of patient care, those with CN involvement were mostly treated as outpatients compared to those with PNS involvement (either isolated or in conjunction with CN involvement), that were more likely to be hospitalized and/or ICU stay (p < .001). Patients with PNS involvement were more likely to receive intravenous immunoglobulins (p = .002) or plasma exchange (p = .002) compared to patients with isolated CN involvement that received mostly glucocorticoids (p< .001). Also, we found no statistically significant

**Table 4. Comparative clinical data according to different types of neurological involvement.**

| Number of patients, N = 328 | | CN involvement N = 102 (%) | PNS involvement N = 161 (%) | CN + PNS involvement N = 65 (%) | p value | Missing data N (%) |
|---|---|---|---|---|---|---|
| Male sex | | 55 (53.9) | 91 (56.5) | 39/63 (61.9) | .63 | 2 (0.6) |
| Age, mean ± SD | | 46.2 ± 17.1 | 52.9 ±15.1 | 51.6 ± 15 | **.003** | 0 |
| Severe COVID-19 | | 25/101 (24.8) | 59/158 (37.3) | 22/63 (34.9) | .1 | 6(1.8 |
| Duration between disease onset and neurological symptoms (days), median (range) | | 8 (0–90) | 13.5 (0–122) | 13.5 (0–42) | .06 | 65 (19.8) |
| Level of patient care | Outpatient | 35/87 (40.2) | 15/115 (13) | 5/60 (8.3) | < .001 | 66 (20.1) |
| | Normal ward | 33/87 (37.9) | 47/115 (40.9) | 28/60 (46.7) | | |
| | ICU | 19/87 (21.8) | 53/115 (46.1) | 27/60 (45) | | |
| Treatment | GC | 45 (44.1) | 12 (7.5) | 16/65 (24.6) | < .001 | 0 |
| | IVIG | 38 (37.3) | 81 (50.3) | 17/65 (26.2) | **.002** | |
| | PLEX | 0 (0) | 19 (11.8) | 5/65 (7.7) | **.002** | |
| | No treatment | 11 (10.8) | 15 (9.3) | 16 (24.6) | **.006** | |
| | Others | 3 (2.9) | 5 (3.1) | 1 (1.5) | .8 | |
| Outcome | Recovered completely | 24/95 (25.3) | 24/123 (19.5) | 12/59 (20.3) | .1 | 51 (15.5) |
| | Mild/moderate sequelae | 52/95 (54.7) | 83/123 (67.5) | 40/59 (67.8) | | |
| | No improvement | 17/95 (17.9) | 10/123 (8.1) | 4/59 (6.8) | | |
| | Non-survivors | 2/95 (2.1) | 6/123 (4.9) | 3/59 (5.1) | | |

Abreviations: CN–cranial nerves, PNS–peripheral nervous system, ICU–intensive care unit, GC–glucocorticoids, IVIG–intravenous immunoglobulins, PLEX–plasma exchange

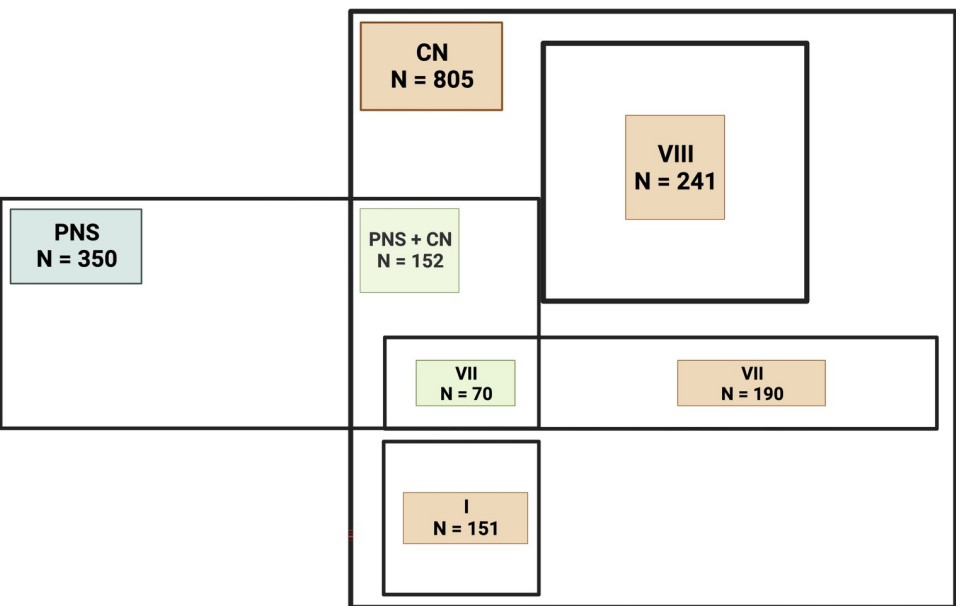

**Fig 2. Peripheral nervous system involvement; PNS–peripheral nervous system, PNS+CN–peripheral nervous system and cranial nerves, I–olfactory nerve, VII–facial nerve, VIII–vestibulo-cochlear nerve.**

differences regarding COVID-19 disease severity among all three groups of patients, with severe forms of disease in less than 40% in all categories. When comparing the groups in Table 4, duration between disease onset and the neurological symptoms was shorter for the CN groups– 8 (0–90) days, versus 13.5 (0–122) and 13. 5 (0–42) respectively (p = .06).

In Fig 2 we represented the overlapping between the PNS and CN involvement emphasizing the most commonly affected CN (I, VII, VIII).

## Discussions

### Mechanisms of neurological involvement in SARS-CoV-2 infection

In this systematic review we identified various peripheral nerve manifestations, ranging from the classic form of GBS to a wide range of CN symptoms. Regarding the mechanisms of the neurological involvement, two main theories have been postulated: the neurotropism and neuroinvasion of the SARS-CoV-2 virus on the one hand, and the dysimmune response to the virus on the other [5–7, 9–13, 15–23, 25–57].

SARS-CoV-2 enters the human cell via the angiotensin-converting enzyme 2 (ACE2) receptor using the spike proteins located on its surface. The ACE2 receptor is present on the surface of neurons, glia, endothelial cells. It is more concentrated in certain areas of the brain such as the brainstem, especially in the cardiovascular and respiratory regulation area, nucleus of the tractus solitarius, periventricular areas, etc. [58].

The pathway for the viral neuroinvasion is considered to be the peripheral nerve terminals. The olfactory nerve route is the most cited, but this issue is still under debate. The nasal epithelium has a high concentration of ACE2 receptors, explaining the anosmia and allowing the virus to enter the pseudounipolar neurons. From here, the virus is transported trans synaptically, through the cribriform plate to the olfactory bulb and to the CNS. Another possible pathway is via the enteric nervous system, as there is proof that SARS-CoV2 can invade the enteric

cells via the highly expressed ACE2 receptors in the small intestine. At this level the virus can be transported to the CNS via both antero- and retro-grade transport [1, 39, 58].

Song et. al. studied the effects of the novel coronavirus on neural tissues of both human and mouse origin, suggesting that the neurological symptoms are a result of direct viral invasion. It appears that the neuroinvasive characteristics of SARS-CoV-2 are different from other neuro-tropic viruses (like rabies, or HSV) as it does not induce a lymphocyte infiltration around the positive viral staining areas [59].

SARS-CoV-2 induces also an important inflammatory response, especially in patients with severe disease, known as the "cytokine storm". As a consequence of the innate immune response, an excessive release of pro-inflammatory cytokines (IL-1, IL-6, IL-8, IL-12, TNF-$\alpha$, interferons, CXCL-10, MCP-1, MIP-1$\alpha$, G-CSF and also GM-CSF) into the bloodstream, will lead to multiple tissue and organ damage (MODS). This inflammatory response may also trigger a disruption in the blood-brain barrier, causing a hematogenous spread of the virus into the CNS. At the same time, the neuro-immune pathway functions bidirectionally, leading to the inflammatory reflex: the afferent neurons of the vagus nerve respond to immune signals in the periphery and the efferent neurons regulate the pro-inflammatory cytokines release and will also orchestrate the systemic inflammatory response [1, 38, 55].

Since the beginning of the vaccination campaign, controversies regarding association between GBS and other neuropathies on one hand, and the SARS-CoV-2 vaccines on the other have arisen. To date, no causal relationship has been clearly established, as there may be a coincidental association. Although not the topic of our review, potential association between SARS-CoV-2 vaccines and different immune mediated neuropathies should be more thoroughly analyzed.

## PNS involvement in SARS-CoV-2 infection

GBS is usually a post infectious, immune-mediated peripheral nervous system disease, caused by a dysimmune response to an infectious agent, but in COVID-19 theories about para-infectious manifestations are emerging. Khan et. al. identified the presence of SARS-CoV-2 RNA in the CSF of a GBS patient, raising questions about the direct involvement of the virus in GBS physiopathology. In the same case series, there is a patient who developed AIDP symptoms one day after COVID-19 symptoms, but PCR for SARS-CoV-2 was negative in the CSF. The same pattern of para-infectious GBS was also observed in other case reports [20, 36, 37]. On the other hand, in a study comparing the COVID-19 associated GBS incidence with the general GBS incidence for the same period a year before the pandemic, no epidemiological and phenotypical association between SARS-CoV-2 and GBS was found [12]. The majority of patients who developed GBS secondary to COVID-19 follows the known disease pattern: ascending neurological deficits (sensorial, motor or both) few weeks after an infection. There are still questions to answer regarding the physiopathology in patients who develop GBS symptoms prior to or immediately after presenting other SARS-CoV-2 manifestations, as the RNA presence in the CSF is a rare finding, even in this category [35]. In a multicentric study from India patients with post infectious (after four weeks) GBS had a better outcome than those with para infectious GBS [9].

All types of PNS manifestations were found isolated in 60% to more than 90% of reported events, rather than associated with CN involvement. Symptoms of dysautonomia are varied, ranging from sudomotor dysfunction to tachycardia and orthostatic hypotension. The mechanism differs, as sudomotor dysfunction is the result of small fiber neuropathy, and tachycardia and orthostatic hypotension could be the result of the nucleus tractus solitarius dysfunction

due to SARS-CoV-2 neuroinvasion. In one case a rare finding of sudomotor dysfunction preceding AMAN was reported [13, 34].

MNM is usually associated with some types of vasculitis, so SARS-CoV-2 may cause MNM by a combined mechanism of dysimmunity and microthrombi of the vasa nervorum [31–33]. A similar cause was hypothesized in a patient who presented brachial plexopathy preceded by purpuric rash on his arm during hospitalization for COVID-19 [30]. He developed the symptoms after at least 12 days since the onset of COVID-19 symptoms, so an immune mediated cause is more plausible.

Most patients with PNS manifestations presented with a non-severe form of COVID-19 and had a good neurological outcome although the majority remained with mild to moderate sequelae [9]. Patients with GBS spectrum were mostly treated with IVIG and plasma exchange [60]. MFS, part of the GBS spectrum involving the CN along with ataxia and hyporeflexia, was also treated with IVIG with good neurological outcome, some patients remaining only with hyporeflexia [9, 12, 61–68]. In one case the symptoms resolved spontaneously [69]. The same spontaneous resolution was found in the case of pharyngo-cervical-brachial form of GBS [23].

Of the two patients with carpal tunnel syndrome, one remained with mild paresthesia in the fifth finger, although both underwent decompressive surgery [24]. The patients with Parsonage Turner syndrome, although they received glucocorticoids and IVIG, remained with significant motor deficits [25, 26]. The patients with MNM remained mostly with fatigability and didn't need any neurological treatment [33]. In a MNM case complicated with Kawasaki disease and myocarditis, the patient had a good outcome but with persistent bilateral foot drop four months later after treatment with rituximab, cyclophosphamide and IVIG [31]. Patients with dysautonomia received no specific treatment [13].

## CN involvement in SARS-CoV-2 infection

The CN were affected either in isolation or associated with the peripheral nerves. The facial nerve was the most affected CN overall and in association with other peripheral neuropathies, but it was still more frequently affected isolated. Whether this is the result of direct nerve invasion, or it is also the result of an inflammatory response, it remains to be settled. Also, a reactivation of herpes simplex virus or varicella-zoster virus cannot be ruled out, as COVID-19 can induce a state of immunosuppression. Codeluppi et.al. observed a 21% raise in peripheral facial nerve palsy in the pandemic compared to the same period of the previous year, as well as younger age of patients [51]. Similar findings were reported by others. [7, 35].

The vestibulocochlear nerve was the second most affected nerve and was mostly found affected in isolation. The most common symptoms were sudden sensory-neural hearing loss (SSNHL), tinnitus and vertigo. The majority of reported cases presented with unilateral involvement, but in one case series bilateral involvement was more frequent. If unilateral SSNHL is usually a sign of direct viral invasion as is the case with other viruses (HSV, CMV), bilateral symptomatology is more prone to being the consequence of a systemic response, such as inflammation or vasculitis. In a study conducted by Dharmarajan et. al. 48% of the patients with audiometric proof of SSNHL had no symptom of SSNHL, suggesting an even larger potential number of COVID-19 patients with subclinical vestibulo-cochlear nerve involvement [4, 10].

The oculomotors, the vagus, glossopharyngeal, accessory and hypoglossal nerves were also more frequently affected isolated rather than associated with other types of peripheral neuropathies. The trigeminal nerve is the only CN that was more often associated with peripheral neuropathies, rather than in isolation. The least affected CN overall was the hypoglossal nerve (XII). In one prospective observational study it was found that the sensory functions of the nerves were more affected than the motor ones [6].

Even though our study did not include patients that presented with isolated anosmia, it was most likely associated with other CN manifestations rather than peripheral neuropathies. In a study measuring the olfactory cleft it was found that larger width and volume of the olfactory cleft was a predisposing factor for anosmia in COVID-19 patients, as it may result in more prompt nasal immune response and facilitate access to the olfactory bulb [8].

Patients with CN involvement had mostly non severe forms of COVID-19 and received glucocorticoids either locally, or systemically, or both, with various results. For example, some patients with SSNHL recovered their hearing partially, while others had mild or no improvement, but early identification and treatment was linked to better results [4, 10, 18]. Isolated facial palsy was also treated with glucocorticoids, with good results. Patients with facial diplegia and GBS received mostly IVIG with symptoms resolution in various degrees. [51, 70–72].

Our quantitative synthesis showed that patients with isolated CN involvement were more likely to be outpatients, while patients with isolated PNS involvement were more prone to staying in the ICU. We should take into consideration that most studies did not mention the reason for the ICU stay–either the severe type of disease, or the neurological involvement, but given the fact that GBS is usually treated in an ICU, this may be a reasonable assumption.

Regarding treatment, patients with isolated CN involvement received mostly glucocorticoids, either systemically, locally, or both, while patients with PNS involvement were more likely to receive IV immunoglobulins or plasma exchange, similar to the standard treatment of these findings secondary to other etiologies [73].

There was no difference in sex distribution between groups, with male patients representing over 50% of patients in all three categories. Although the difference was not significant, patients with CN involvement had a shorter period between disease onset and neurological symptoms than the other two groups, suggesting a possible increased susceptibility of the CN to either direct neuroinvasion or to the inflammatory mechanism. The outcome was mostly favorable, the majority of patients in all three categories remaining with mild and moderate neurologic sequelae. A small number of patients died, but it's difficult to say if the cause was the neurological involvement or the evolution of the COVID-19 disease. All three categories of patients were mostly associated with a non-severe type of COVID-19 disease. Patients with CN involvement were more likely to have a complete recovery compared to those with PNS involvement (isolated or PNS + CN involvement) therefore, the outcome could be related more to the neurologic involvement.

Our evidence synthesis of CN and PNS involvement by SARS-CoV-2 infection provided a comprehensive description of both the diverse clinical scenarios and associated pathophysiological mechanisms. The strengths of our study include also a review of potential treatments and associated outcomes of patients with CN and PNS manifestations associated with COVID-19, as a result of a secondary performed analysis between different types of neurological involvement.

There are several important limitations to this review. First of all, most articles were case reports and case series, and only few prospective studies were identified. Second, there was a significant number of cases of unclassified GBS as the access to diagnostic tools was limited during the pandemic. Also, neurological findings in patients with severe COVID-19 may not have been sufficiently reported because of focusing mainly on COVID-19 and death before any neurological investigations could be made, leading to an underestimation. On the other hand, outpatients with mild neurological symptoms and mild COVID-19 may not have presented for care or data may not have been reported.

Many patients were lost to follow up, therefore the data available for the outcome may be biased.

However, at this time point, insufficient data is available on this topic and further research needs to be assessed in order to generate clear clinical correlations and therapeutic decision-making strategies.

## Conclusions

In conclusion, PNS involvement in COVID-19 may be an important cause of hospitalization and post COVID-19 sequelae, increasing the burden on the healthcare systems, even though they are usually associated with mild and moderate forms of disease. Additional studies following consecutive patients with various severities of COVID 19 forms are needed in order to evaluate the incidence of neurological events and to identify the risk factors. Randomized clinical trials are also needed to assess different interventions for the management of neurological manifestations.

## Supporting information

**S1 Fig. PRISMA 2009 checklist.**
(TIF)

**S1 Table. Characteristics of patients included in the qualitative analysis.**
(DOCX)

## Author Contributions

**Conceptualization:** Andreea-Raluca Hanganu, Alexandru Constantin, Cristian Băicuș, Adriana Hristea.

**Data curation:** Andreea-Raluca Hanganu, Alexandru Constantin.

**Formal analysis:** Cristian-Mihail Niculae, Ioana Diana Olaru.

**Investigation:** Andreea-Raluca Hanganu, Elena-Sonia Moise.

**Methodology:** Andreea-Raluca Hanganu, Alexandru Constantin, Cristian-Mihail Niculae, Cristian Băicuș, Adriana Hristea.

**Project administration:** Adriana Hristea.

**Supervision:** Cristian Băicuș, Adriana Hristea.

**Validation:** Ioana Diana Olaru.

**Visualization:** Andreea-Raluca Hanganu.

**Writing – original draft:** Andreea-Raluca Hanganu, Alexandru Constantin, Elena-Sonia Moise, Cristian-Mihail Niculae.

**Writing – review & editing:** Ioana Diana Olaru, Cristian Băicuș, Adriana Hristea.

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
