## [Decision Letter · Decision Letter 0]

20 Feb 2023

PONE-D-22-25367Peripheral nervous system involvement associated with COVID-19. A systematic review of literaturePLOS ONE

Dear Dr. Hanganu,

Thank you for submitting your manuscript to PLOS ONE. After careful consideration, we feel that it has merit but does not fully meet PLOS ONE’s publication criteria as it currently stands. Therefore, we invite you to submit a revised version of the manuscript that addresses the points raised during the review process.

ACADEMIC EDITOR: Although several articles have been published on this topic, your analysis is well structured. Please address the comments below. ==============================

We look forward to receiving your revised manuscript.

Kind regards,

Victor Daniel Miron

Academic Editor

PLOS ONE

Additional Editor Comments:

The topic of the manuscript is of interest and although numerous case series/studies and meta-analyses have been published on this topic to date, this manuscript is clearly written but requires minor clarification:

- "SARS-CoV-2 is a respiratory virus," - I think here it should be rephrased. SARS-CoV-2 is a respiratory virus spread, but more recent studies show that the primary tropism of SARS-CoV-2 is not limited to the respiratory tract.

- In the introduction, I recommend deleting the sentence, as since the submission of this manuscript several reviews on this topic have been published: "By contrast, cranial nerve (CN) involvement and other PNS manifestations have not been extensively reviewed.".

- The methods are well described.

- Data on the time of onset of neurological manifestations in relation to the onset of the first symptoms of COVID-19 should be included. In the methods you state that you have taken into account these data regarding the time of onset. Is there any correlation between specific neurological manifestations and the time of onset?

- The discussion is well written.

- The abstract should be structured according to journal recommendations.

Reviewers' comments:

Reviewer's Responses to Questions

**Comments to the Author**

1. Is the manuscript technically sound, and do the data support the conclusions?

Reviewer #1: Yes

Reviewer #2: Partly

2. Has the statistical analysis been performed appropriately and rigorously? 

Reviewer #1: I Don't Know

Reviewer #2: Yes

3. Have the authors made all data underlying the findings in their manuscript fully available?

Reviewer #1: Yes

Reviewer #2: Yes

4. Is the manuscript presented in an intelligible fashion and written in standard English?

Reviewer #1: Yes

Reviewer #2: No

5. Review Comments to the Author

Reviewer #1: In this systematic review, the authors aim to investigate the characteristics, management and outcomes of patients with PNS manifestations. In my opinion, the authors achieved their objectives in this paper, so I recommend acceptance of this paper for publication.

Reviewer #2: The manuscript by Andreea-Raluca Hanganu et al. entitled “Peripheral nervous system involvement associated with COVID-19. A systematic review of literature” aimed to investigate the Characteristics, management and outcomes of patients with PNS, including the types and severity of cranial nerves (CN) involvement. The paper is well written and the discussion is honest in relation to the results. The abstract summarizes the general significance of the manuscript and the article leads some evidence to such point, but there are some major issues need to be addressed to improve the significance of the manuscript:

-Firstly, articles reporting neuropathies in patients vaccinated for SARS-CoV-2 within the previous three months in whom the neurological findings could have been attributed to vaccination were excluded. However, it would be interesting to include a paragraph on the vaccination/neuropathies relationship in the discussion.

- The authors should include some figures to make the text easier to read.

-Finally, the bibliography should be expanded; accordingly, this article should be cited:

Visco V, Vitale C, Rispoli A, Izzo C, Virtuoso N, Ferruzzi GJ, Santopietro M, Melfi A, Rusciano MR, Maglio A, Di Pietro P, Carrizzo A, Galasso G, Vatrella A, Vecchione C, Ciccarelli M. Post-COVID-19 Syndrome: Involvement and Interactions between Respiratory, Cardiovascular and Nervous Systems. J Clin Med. 2022 Jan 20;11(3):524. doi: 10.3390/jcm11030524. PMID: 35159974; PMCID: PMC8836767.

6. PLOS authors have the option to publish the peer review history of their article (what does this mean?). If published, this will include your full peer review and any attached files.

Reviewer #1: No

Reviewer #2: No

---

## [Author Response · Author response to Decision Letter 0]

15 Mar 2023

Dear editor and reviewers,

Thank you for your valuable comments, allowing us to improve our manuscript. 

Please find below the responses to the raised points:

1. Please include captions for your Supporting Information files at the end of your manuscript, and update any in-text citations to match accordingly: - see row 428

2. Please review your reference list to ensure that it is complete and correct.

As suggested, I revised the reference list and there is an article (doi: 10.7759/cureus.14391) that has been retracted because of fake peer review (doi:10.7759/cureus.8992). Thus, we redid the statistics but with no change in the final results. In the original manuscript the article was cited at no. 40. We also eliminated it from the Supplement Table 1.

3. "SARS-CoV-2 is a respiratory virus," - I think here it should be rephrased. SARS-CoV-2 is a respiratory virus spread, but more recent studies show that the primary tropism of SARS-CoV-2 is not limited to the respiratory tract – rephrased, see row 48

4. In the introduction, I recommend deleting the sentence, as since the submission of this manuscript several reviews on this topic have been published: "By contrast, cranial nerve (CN) involvement and other PNS manifestations have not been extensively reviewed." – deleted, see row 60

5. In the methods you state that you have taken into account these data regarding the time of onset. Is there any correlation between specific neurological manifestations and the time of onset? –

The results regarding the duration between disease onset and neurological manifestations were added in table no 4, row 237 and row 386. No statistical association was found.

6. The abstract should be structured according to journal recommendations.

We structured the abstract according to the recommendations: we described the main objective of the study, how the study was performed without details, summarized the most important results and their significance, we did not exceed 300 words and did not use citations. 

We used abbreviations (which are not recommended but are not forbidden) because of the frequent use of the terms “cranial nerves” and “peripheral nervous system”. Not using abbreviations would have significantly limited our word count to present our results. 

Did we understand correctly that you suggested deleting the subheadings? 

7. However, it would be interesting to include a paragraph on the vaccination/ neuropathies relationship in the discussion – see row 282

8. The authors should include some figures to make the text easier to read – see Figure 2, row 242

9. Finally, the bibliography should be expanded; accordingly, this article should be cited: Visco V, Vitale C, Rispoli A, Izzo C, Virtuoso N, Ferruzzi GJ, Santopietro M, Melfi A, Rusciano MR, Maglio A, Di Pietro P, Carrizzo A, Galasso G, Vatrella A, Vecchione C, Ciccarelli M. Post-COVID-19 Syndrome: Involvement and Interactions between Respiratory, Cardiovascular and Nervous Systems. J Clin Med. 2022 Jan 20;11(3):524. doi: 10.3390/jcm11030524. PMID: 35159974; PMCID: PMC8836767 – see reference no 61

In addition, we made some changes at the PRISMA flow chart regarding the number of articles included in the qualitative synthesis because of a transcription error, and in the quantitative analysis because of the retracted paper.

 Best wishes,

Andreea Hanganu

---

## [Editor Report · Decision Letter 1]

20 Mar 2023

Peripheral nervous system involvement associated with COVID-19. A systematic review of literature

PONE-D-22-25367R1

Dear Dr. Hanganu,

We’re pleased to inform you that your manuscript has been judged scientifically suitable for publication and will be formally accepted for publication once it meets all outstanding technical requirements.

Kind regards,

Victor Daniel Miron

Academic Editor

PLOS ONE

---

## [Editor Report · Acceptance letter]

28 Mar 2023

PONE-D-22-25367R1 

Peripheral nervous system involvement associated with COVID-19. A systematic review of literature 

Dear Dr. Hanganu:

I'm pleased to inform you that your manuscript has been deemed suitable for publication in PLOS ONE. Congratulations! Your manuscript is now with our production department. 

Kind regards, 

on behalf of

Dr. Victor Daniel Miron 

Academic Editor

PLOS ONE